# Delivery of Mesenchymal Stem Cells during Hypothermic Machine Perfusion in a Translational Kidney Perfusion Study

**DOI:** 10.3390/ijms25095038

**Published:** 2024-05-05

**Authors:** Natalie Vallant, Nienke Wolfhagen, Bynvant Sandhu, Karim Hamaoui, Vassilios Papalois

**Affiliations:** Department of Surgery and Cancer, Imperial College London, London SW7 2AZ, UK; n.vallant@imperial.ac.uk (N.V.); nienkewolfhagen@gmail.com (N.W.);

**Keywords:** ischemia reperfusion injury, Mesenchymal Stem Cells, kidney transplantation

## Abstract

In transplantation, hypothermic machine perfusion (HMP) has been shown to be superior to static cold storage (SCS) in terms of functional outcomes. Ex vivo machine perfusion offers the possibility to deliver drugs or other active substances, such as Mesenchymal Stem Cells (MSCs), directly into an organ without affecting the recipient. MSCs are multipotent, self-renewing cells with tissue-repair capacities, and their application to ameliorate ischemia- reperfusion injury (IRI) is being investigated in several preclinical and clinical studies. The aim of this study was to introduce MSCs into a translational model of hypothermic machine perfusion and to test the efficiency and feasibility of this method. Methods: three rodent kidneys, six porcine kidneys and three human kidneys underwent HMP with 1–5 × 10^6^ labelled MSCs within respective perfusates. Only porcine kidneys were compared to a control group of 6 kidneys undergoing HMP without MSCs, followed by mimicked reperfusion with whole blood at 37 °C for 2 h for all 12 kidneys. Reperfusion perfusate samples were analyzed for levels of NGAL and IL-β by ELISA. Functional parameters, including urinary output, oxygen consumption and creatinine clearance, were compared and found to be similar between the MSC treatment group and the control group in the porcine model. IL-1β levels were higher in perfusate and urine samples in the MSC group, with a median of 285.3 ng/mL (IQR 224.3–407.8 ng/mL) vs. 209.2 ng/mL (IQR 174.9–220.1), *p* = 0.51 and 105.3 ng/mL (IQR 71.03–164.7 ng/mL) vs. 307.7 ng/mL (IQR 190.9–349.6 ng/mL), *p* = 0.16, respectively. MSCs could be traced within the kidneys in all models using widefield microscopy after HMP. The application of Mesenchymal Stem Cells in an ex vivo hypothermic machine perfusion setting is feasible, and MSCs can be delivered into the kidney grafts during HMP. Functional parameters during mimicked reperfusion were not altered in treated kidney grafts. Changes in levels of IL-1β suggest that MSCs might have an effect on the kidney grafts, and whether this leads to a positive or a negative outcome on IRI in transplantation needs to be determined in further experiments.

## 1. Introduction

Kidney transplantation remains the only curative treatment for patients with end-stage renal failure. In times of organ shortage, new strategies for graft preservation have been investigated with an aim to not only preserve, but to optimize organs prior to transplantation. Hypothermic machine perfusion (HMP) has been demonstrated to reduce the rates of delayed graft function and to improve graft survival in the first year of transplantation, when compared to static cold storage (SCS). This effect was observed for both kidneys from braindead donors (DBD) and kidneys from deceased donors (DCD) [1,2,3,4]. HMP has been widely adopted into routine practice, with commercial devices facilitating the option of organ perfusion even during transport from one center to another.

A next step to perfectly exploit the benefits of the machine perfusion of organs prior to transplantation is the potential of preconditioning grafts ex vivo via the use of different perfusates and/or via the introduction of cytotopic anticoagulants [5,6] or Mesenchymal Stem Cells (MSC) into the circuit. MSCs are multipotent, self-renewing cells which can be isolated from various types of tissue, such as bone marrow, adipose tissue and many others [7]. In addition to their capacity to differentiate into osteoblasts, adipocytes and chondrocytes [8], MSCs possess immunomodulatory and tissue-regenerative capacities [9,10]. The cells have been used as an experimental therapeutic agent in graft-versus-host disease (GVHD) [11] and Crohn’s disease [12], and reported safety properties of MSCs in preliminary data from these clinical trials have initiated interest on their use in solid organ transplantation (SOT). Recent analyses suggested that MSCs could potentially be used to treat ischemia-reperfusion injury (IRI), acute rejection and even to precondition organs and recipients in order to attenuate or even avoid those unwanted side effects [13,14,15]. So far, research has mainly focused on systemic, intravenous infusion of MSCs to recipients before or around the time of transplantation. Studies suggest that systemically delivered MSCs home in on sites of injury in a time-dependent manner but that systemic delivery is limited by entrapment of the cells in the lungs [16,17]. In order for MSCs to have the best effect, direct infusion into the target organ might be the solution. HMP could be a tool to facilitate engraftment of the cells into an organ before transplantation. As described recently by Merel Pool et al., MSCs delivered via normothermic machine perfusion (NMP) in a porcine ex vivo perfusion model could be traced within the glomeruli [18]. As discussed in previous work by us, contrary to NMP, for which no commercially available kidney perfusion device is available, a sophisticated setup is required, and highly trained medical and engineering personnel are required, HMP poses a more straightforward method of organ preservation, which can easily be performed by a wide spectrum of trained individuals. HMP solutions are commercially available and ready to use after storage in the fridge.

The purpose of this study was to investigate whether the administration of MSCs in a translational HMP model is feasible, whether MSCs are viable at low temperatures during HMP, and whether labelled MSCs can be traced within the kidneys after HMP. We furthermore wanted to investigate whether infused MSCs had any potential negative effects on the kidney grafts in the immediate reperfusion phase. We used a translational model, tracing GFP-positive MSCs in rodent kidneys, investigating the infusion of double-labelled cells into a bigger cohort of porcine kidneys undergoing HMP prior to a 2 h reperfusion period and tracing MSCs in human kidneys after HMP.

## 2. Results

### 2.1. Mesenchymal Stem Cells in HMP in Rodent Kidneys

Three rodent kidneys were hypothermically perfused ex vivo, with a perfusate (50 mL of MPS) containing 3 × 10^6^ green-fluorescent MSCs (n = 3) for one hour.

We were able to locate green-fluorescent MSCs within the rodent kidneys, using widefield microscopy. Figure 1a shows images of a frozen kidney section at a 20× magnification. Covered in DAPI, nuclei are visible in blue. Mesenchymal Stem Cells in bright green are clearly visible, mainly within the glomeruli of the kidney. Figure 1b shows one field of that section further magnified, at 40× magnification, and green-fluorescent MSCs are clearly visible within the glomeruli of the rodent kidney (red arrows).

### 2.2. Mesenchymal Stem Cells in a Porcine HMP Model

#### 2.2.1. Perfusion Parameters during HMP

During 4 h of HMP versus HMP + MSC, we did not observe any significant differences in regard to perfusate flow rates or (intrarenal resistance indices) RRIs between the groups, as shown in Figure 2a,b. RRIs at all time points were slightly higher in the HMP + MSC group, with mean values of 0.99 ± 0.62 mmHg/mL/min in the HMP + MSC group vs. 0.82 ± 0.5 mmHg/mL/min in the HMP group (*p* = 0.86), and perfusate flow rates were slightly lower, at 16.97 ± 4.37mL/min/100 g in the HMP + MSC group vs. 18.21 ± 5.59 mL/min/100 g in the HMP group (*p* = 0.65).

#### 2.2.2. Perfusion Parameters during Normothermic Reperfusion

During the normothermic reperfusion of porcine kidney pairs, similar physiological parameters were observed for both the MSC-treated and the control groups, as shown in Figure 3a–d. We measured RRIs of 0.46 ± 0.16 mmHg/mL/min in the HMP group vs. 0.45 ± 0.17 mmHg/mL/min in the HMP + MSC group (*p* = 0.94, Figure 3a). Perfusate flow rates were also similar with mean values of 51.77 ± 13.79 mL/min/100 g in the former vs. 51.33 ± 11.42 mL/min/100 g in the latter group (*p* = 0.96, Figure 3b). Mean oxygen consumption rates were 25.48 ± 10.17 mL/min/100 g in the HMP group vs. 24.97 ± 10.72 mL/min/100 g in the HMP + MSC group (*p* = 0.92, Figure 3c), and mean urinary output rates were 4.25 ± 1.87 mL/min in the former and 3.72 ± 1.58 mL/min in the latter group (*p* = 0.77, Figure 3d).

#### 2.2.3. Perfusate Analyses during Normothermic Reperfusion

The creatinine clearance and fractional sodium excretion rates were similar in both groups with a mean creatinine clearance of 4.57 ± 0.75 mmol/min in the HMP group vs. 4.25 ± 0.78 mmol/min in the HMP + MSC group (*p* = 0.89). The mean fractional sodium excretion rate was 0.79 ± 0.21% in the former vs. 0.81 ± 0.3% in the latter group (*p* = 0.87).

An analysis of perfusate and urine samples for Interleukin-1β (IL-1β) and neutrophil gelatinase-associated lipocalin (NGAL) by ELISA revealed a trend towards higher concentrations of IL-1β in perfusate and urine samples of MSC-treated kidneys, especially towards the end of the reperfusion phase, as demonstrated in Figure 4a,b, with mean concentrations of 225.8 ± 83.92 pg/mL vs. 151.8 ± 7.98 pg/mL (*p* = 0.51) for perfusate samples (Figure 4a) and 244.2 ± 70.65 pg/mL vs. 132.7 ± 49.5 pg/mL (*p* = 0.15) for urine samples (Figure 4b). The mean values of NGAL perfusate concentrations were similar in both groups, with a mean of 121.0 ± 25.81 pg/mL in the HMP group vs. 129.8 ± 37.56 pg/mL in the HMP + MSC group (*p* = 0.91, Figure 4c). The mean urinary NGAL concentrations were higher, but not significantly, in the HMP group, at 212.6 ± 133 pg/mL, and at 137.5 ± 60.4 pg/mL in the HMP + MSC group (*p* = 0.23, Figure 4d).

#### 2.2.4. Widefield Microscopy Visualization of MSCs

In the pHMP + MSC group, single MSCs could be traced within the glomeruli and tubules of all porcine kidneys, except from the groups in which the smallest number of MSCs, 1 × 10^6^ cells, were infused. MSCs were detected after the HMP phase, as well as after the reperfusion phase. Figure 5 shows photographs of microscopic images taken using widefield microscopy, with a negative control (Figure 5a), cells within the glomeruli (Figure 5b) and within the tubuli (Figure 5c).

#### 2.2.5. Immunohistochemistry

Immunohistochemical staining of porcine samples against human major histocompatibility (MHC) class 1 (MHC-1) confirmed the presence of MSCs within porcine kidneys, both after the HMP phase and in the reperfusion phase. Figure 6 shows photographs of histology slides after immunohistochemical staining and counterstaining with PAS at a 20× magnification. Figure 6a shows the section of a kidney in the HMP group as a negative control. Figure 6b shows a human skin sample, which is known to contain high amounts of MHC-1, stained in brown. Brown-stained human MSCs were detected within the glomeruli (Figure 6c arrows), as well as within tubuli (Figure 6d) of porcine kidneys. Immunohistochemical staining revealed that delivered cells seem to not just be stuck within the organs, but, in the case of the tubules, they are integrated into the endothelium, as shown in Figure 6d (arrows).

### 2.3. Mesenchymal Stem Cells in a Human HMP Model

The same perfusion protocol for HMP was applied as was used for the porcine organs, with the aim to investigate whether it was possible to introduce MSCs into a human organ ex vivo. Three consecutive single kidneys were used for this purpose, without any comparative experiment. The kidneys underwent HMP, with of 1 × 10^6^, 3 × 10^6^ and 5 × 10^6^ double-labelled MSCs added to the circuit, respectively. For every single kidney undergoing perfusion with MSCs, we kept the cold ischemia time (CIT) at exactly 24 h by either keeping them on ice or declining offers of kidneys for which a longer CIT was expected. Within human kidneys, we were able to detect single double-labelled cells within glomeruli and tubuli (Figure 7a,b). Also, in the case of the perfusion of human kidneys, only with at least 3 × 10^6^ cells within the perfusate were we able to trace them within an organ using widefield microscopy.

## 3. Discussion

Currently, many studies are focusing on the administration of Mesenchymal Stem Cells as a potential agent to alleviate IRI in transplantation. MSCs have been shown to have anti-inflammatory and regenerative capacities and, furthermore, to home in on sites of injury [19,20,21,22]. So far, intravenous infusion has been the route of administration in most clinical studies, including renal transplantation, and has been shown to be safe [23]. However, one of the major problems with that has been that MSCs get trapped within capillaries of the lungs before reaching the target organ [16]. Furthermore, the efficacy of delivered MSCs has been found to be dependent on the timing of administration peri-transplantation. Liu et al. [24] described that MSC treatment for acute kidney injury was most effective when applied prior to the development of an inflammatory microenvironment, and in their study, urinary NGAL was described as a potent marker to estimate when to infuse MSCs because it correlated well with the inflammatory state of kidneys. Also, in animal studies, MSC treatment prior to renal ischemia reperfusion injury was demonstrated to have protective effects [25].

A combination of ex vivo HMP and delivery of MSCs into solid organs has not been described in the literature yet. In this study, we focused on this potential new pathway for organ preconditioning ex vivo in a translational preclinical model. Starting in a rat kidney model, we had the opportunity to test the principle with the use of MSCs extracted from transgenic Wistar Kyoto (WKY) GFP rats, which ubiquitously carry green-fluorescent protein [26]. The fluorescent cells were perfused into kidneys from wild-type animals, and therefore, it was very easy to identify infused cells within treated kidneys. The hypothermic machine perfusion setup for this model was very simple and consisted only of the basic components for HMP. MSCs could be identified within the kidneys after perfusion, and it seems like a majority of the cells were located within the glomeruli (Figure 1a,b). Whether this is the effect of the cells ‘homing’ in on sites of injury, or simply getting trapped in the capillary bed of glomeruli due to their size needs to be determined. There are no clear guidelines on or recommendations for an ideal cell number when introducing MSCs as a treatment for IRI into an isolated organ. Our chosen number, 1–5 × 10^6^, was a result of our findings in the literature, where similar numbers of cells had been used for systemic infusions to target damaged organs.

In the next step, six consecutive porcine kidney pairs underwent HMP, with one respective kidney of a pair receiving MSC treatment on the machine and the other kidney serving as control. Allocations to each arm in regard to the side were performed blindly and randomly. We believe that having organs from the same respective donor animal serve as a control for each experiment can be very useful in order to account for otherwise occurring differences between individual animals.

During 4 h of HMP, only minor differences between treated and non-treated kidneys in terms of functional parameters were observed, e.g., IRR and flow rate. The graphs for each group look almost identical (shown in Figure 2a,b). Also, during a mimicked reperfusion phase, no differences in physiological parameters between the two groups were found (Figure 3a–d). Interestingly, we observed higher levels of the kidney injury markers NGAL and the inflammatory marker IL-1b in urine samples of kidneys perfused with MSCs during the 2 h reperfusion phase; however the differences were not statistically significant (Figure 4b,d). This highlights that the cells might alter the inflammatory environment within these kidneys. The question will be whether this will have a protective effect or not, and this can only be answered by conducting more research on this topic, e.g., translation into in vivo studies. This project’s purpose was foremost to show that MSCs can be delivered into kidney grafts during HMP and that inflammatory profiles were not significantly altered. Out of the six kidneys undergoing HMP with MSC treatment, two kidneys were perfused with 1 × 10^6^ cells, two kidneys received 3 × 10^6^ MSCs and two kidneys were infused with 5 × 10^6^ cells. We found that infusion of 1 × 10^6^ MSCs resulted in difficulties in regard to tracing the cells within the kidneys histologically. We think that there would still be cells within those kidneys, but that it was simply more difficult to capture them in small punch biopsy samples which were then cut into microscopically thin sections. Our porcine model demonstrated that infusion of MSCs during HMP did not result in any immediate negative effects upon a 2 h period of reperfusion. In our opinion, this serves as an important proof that the pre-treatment of kidney grafts with MSCs in a HMP setting is feasible. It would of course be interesting to investigate the effects of delivered cells in a transplantation to ultimately test for safety of this method. Recently, an auto-transplantation study in a porcine model also demonstrated the successful delivery of MSCs in an NMP model, and no negative side effects in the post-transplant period were detected [27]. Due to the fact that HMP is easier to implement in clinical practice and might even result in better outcomes in a porcine model [28], our findings are an important step towards future research within this field. Using human MSCs for porcine organs gave us the advantage of not only tracing cells via the fluorescent cell dye but also confirming the findings with immunohistochemical staining against human proteins. Staining against human MHC class-1 was specific for the human cells only. It was interesting to find that MSCs were detectable within glomeruli, but also, on some of the pictures, they seemed to be incorporated into the endothelial cell layer of the tubules. Indeed, several mechanisms of the acceleration of regenerative capacities of MSCs have been identified, ranging from wound healing and angiogenesis by secretion of pro-angiogenic factors to the differentiation into endothelial cells and/or pericytes [29].

In the last step of this translational study, we treated three consecutive kidneys which had been rejected from clinical transplantation for mentioned reasons with double-labelled MSCs. The main aim was to proof the principle of successful delivery and tracking of the cells within the kidneys after HMP. As shown in Figure 6a,b, we were again able to detect the cells within glomeruli and tubuli of the kidneys using widefield microscopy. Again, only in the case of delivery of >3 × 10^6^ MSCs was it possible to find the cells. Due to the fact that the kidneys used were single kidneys of different donors, we did not conduct a comparative study with a focus on perfusion parameters, and due to the lack of autologous whole blood, we did not mimic reperfusion for these organs. Perfusion parameters during the 4 h periods of HMP were, however, stable and comparable to kidneys undergoing HMP without MSC infusion. For experiments using human MSCs, we decided to label our MSCs with cell dye rather than to transfect them with GFP with an aim to avoid losing cells within the process and furthermore, to affect their phenotype as little as possible. Labelling the cells enabled us to definitely distinguish delivered MSCs from cells of kidney donor origin. Cells under hypothermic conditions are supposed to be metabolically inactive. However, the fact that the cells carried the cytosolic and the cell membrane dye was a sign of viability as apoptotic cells would most likely have lysed. Unfortunately, the quantification of delivered cells into the kidneys was technically not possible using thin sections for microscopy.

We are the first group to demonstrate that it is possible to deliver MSCs into organs by ex vivo hypothermic machine perfusion and to trace the cells within the organs after doing so. Another question will be whether potential effects would be triggered by direct cell–cell interactions in an indirect way, for example by secretion of chemokines or microvesicles [30].

## 4. Materials and Methods

All studies were performed with ethical approval and approval under the Animal Scientific Procedures Act (1986). Experiments were performed under Project License Number PB1C4696D, granted by the Home Office, London, UK.

### 4.1. Study Groups

**Group rHMP + MSC:** rodent kidneys undergoing 1 h of HMP + 3 × 10^6^ MSCs (n = 3).**Group pHMP:** porcine kidneys after 25 min of WIT, 24 h CSS, followed by 4 h of HMP and 2 h of normothermic reperfusion with autologous porcine whole blood (n = 6).**Group pHMP + MSC:** porcine kidneys after 25 min of WIT, 24 h CSS, followed by 4 h of HMP + 1–5 × 10^6^ human MSCs and 2 h of normothermic reperfusion with autologous porcine whole blood (n = 6).**Group hHMP + MSC:** human kidneys after 24 h of CSS and 4 h of HMP + 1 × 10^6^ (n = 1), 3 × 10^6^ (n = 1), or 5 × 10^6^ human MSCs (n = 1).

### 4.2. Mesenchymal Stem Cells

#### 4.2.1. Rat Mesenchymal Stem Cells

Rodent bone marrow-derived Mesenchymal Stem Cells (BM-MSCs) were extracted from femurs and tibias of male Wistar Kyoto (WKY) rats transgenic for the expression of green fluorescent protein (WKY-GFP) by flushing out the bone marrow with cold PBS and incubation of the resulting cell population in MSC growth medium (Mesencult™, Stemcell Technologies, Vancouver, BC, Canada). MSC phenotype of the green, fluorescent cells was confirmed by differentiation into adipocytes, osteocytes, and chondrocytes and by flow cytometry, confirming the presence of clusters of differentiation (CD) CD29, CD44 and CD90; and the absence of CD45 and CD34. The following antibodies were used:CD29: PE anti-mouse/rat CD29 (clone: HMβ1-1, Bio Legend Cat. 102207);CD34: Alexa Fluor 647 anti-mouse CD34 (clone: ICO115, Novus Biologicals Cat. NBP2-33076AF647);CD44: RPE mouse anti-rat CD44 (clone OX-50, Bio Rad Cat. MCA643PE);CD45: Alexa Fluor 647 anti-rat CD45 (clone: OX-1, BioLegend Cat. 202212);CD90: PerCP anti-rat CD90/mouse, CD90.1 (Thy-1.1; clone: OX-7, BioLegend Cat. 202512; Lot: B171081).

Cells were expanded in culture (37 °C, 5% CO_2_, saturating humidity) and split when a confluence of 80% was reached. For perfusion experiments, cells of passages P3–P9 were harvested from the flasks by trypsinization, counted, checked under the microscope for fluorescence and viability and finally resuspended in 5 mL of fresh cell culture medium. The suspension was kept in a Falcon tube on ice until it was injected into the perfusion circuit for HMP.

#### 4.2.2. Human Mesenchymal Stem Cells

One cryovial containing 5 × 10^6^ bone marrow-derived human MSCs was kindly donated to us by our collaborators, Prof. Dazzi et al., from King’s College. MSC phenotype was confirmed in their laboratory by quality testing, and the permission to use these cells for research was obtained. Cells were expanded in culture as described above for rodent MSCs. Cells of passages P3–P9 were used for perfusion experiments, fresh from culture, as described above for rodent MSCs.

Medium for MSC expansion:

PLT lysate 5% + MEMalfa, consisting of aMEM, GlutaMAX™, no nucleosides, Invitrogen (Waltham, MA, USA), Cat. Number 32561-094; and PLT lysate, supplier, Cook; product name, PL-S-100; Cat. Number G3521.

#### 4.2.3. Double-Labelling of Human Mesenchymal Stem Cells

In culture, human MSCs were double-labelled using PKH67 fluorescent cell dye (Sigma-Aldrich, St. Louis, MO, USA) for green fluorescence (excitation at 490 nm, emission at 502 nm), and Qdot^®^ (molecular probes^®^, life technologies™ (Waltham, MA, USA)) nano crystals for red fluorescence (excitation at 405–615 nm, emission at 655 nm), according to manufacturer’s instructions. The double-labelling resulted in MSCs featuring a green cell membrane and red particles within the cytosol.

### 4.3. Organ Retrieval

#### 4.3.1. Rat Kidneys

Rodent kidneys (n = 3) were retrieved from male wild-type Wistar Kyoto (WKY) rats weighing 500–600 g, under general anesthesia via midline laparotomy. Left kidneys were retrieved after ligation of the ureter, the aorta and vena cava towards proximal. Immediately after retrieval, the donor animal was sacrificed by exsanguination and according to the Schedule 1 protocol. Ex vivo, the distal aorta was identified, and a 24 G cannula was inserted and held in place with a 4-0 Vicryl tie. Perfusion was started, and leaking aortal side branches were tied with 5-0 Vicryl ties.

#### 4.3.2. Porcine Kidneys

Pairs of porcine kidneys were retrieved from adult female landrace pigs (70–90 kg) following euthanasia at a local abattoir. The warm ischemia time (WIT) was kept at 25 min. Kidneys were flushed with 500 mL of Soltran via the renal artery at hydrostatic pressure of 100 cm H_2_O before being placed on ice in Soltran solution and transported to the laboratory. Kidneys were stored on ice for 24 h before either undergoing 4 h of HMP, or HMP with Mesenchymal Stem Cells on separate RM3 (Waters Medical Systems (Rochester, MN, USA)) machines. For each retrieved pair, one kidney was allocated to the treatment arm, whereas the other kidney served as the control, respectively. Thereafter, kidneys were reperfused on one RM3 machine, with porcine autologous whole blood, for 2 h, at a physiological temperature (37 °C).

#### 4.3.3. Human Kidneys

Three single human kidneys deemed unsuitable for transplantation were obtained and transported to the laboratory. Kidneys then underwent HMP for 4 h, with double-labelled MSCs added to the circuit. The first kidney was from a 49-year-old female DCD donor, and upon arrival at the laboratory, it had undergone a warm ischemia time (WIT) of 20 min, followed by a cold ischemia time (CIT) of 29 h. Reason for decline from clinical transplantation was a neuroendocrine tumor found within the pancreas. This kidney underwent HMP with 1 × 10^6^ double-labelled human MSCs. The second kidney was obtained from a 55-year-old male donor after cardiac death (DCD); the reason for decline for clinical transplantation was ‘patchy perfusion’ and the WIT was 23 min, followed by a CIT of 24 h. This kidney underwent HMP with 3 × 10^6^ double-labelled human MSCs. The third kidney was retrieved from a 77-year-old female donor after brain death (DBD); the reason for decline was that no suitable recipient was found, and the CIT was 28.5 h at the time of arrival at the laboratory. This kidney underwent HMP with 5 × 10^6^ double-labelled human MSCs.

### 4.4. Perfusates

All hypothermic machine perfusion studies were performed using 500 mL of Machine Perfusion Solution (MPS = modified UW; Bridge to Life (Columbia, SC, USA)) at 4 °C. MSCs (in respective numbers) in 5 mL fresh culture medium were added to the circuit via the arterial reservoir with a silicone valve for injection of substances.

At the time of retrieval of porcine organs, autologous whole blood was collected into a flask containing 20,000 IU of Heparin and 750 mg of Cefuroxime. For reperfusion studies in porcine organs, 500 mL of whole blood was diluted 1:1 with 500 mL of 0.9% saline (Baxter (Deerfield, IL, USA)), resulting in 1 L of perfusate for the reperfusion phase. Then, 0.2 g of creatinine and 5000 IU of Heparin were added. A total of 600 mL of the perfusate was used to prime the circuit and start WBNR, and the remaining 400 mL was used to replace insensitive losses (e.g., urinary output of the kidneys).

### 4.5. Machine Perfusion

#### 4.5.1. HMP of Rodent Kidneys

Three consecutively retrieved left rodent kidneys were hypothermically perfused for 1 h at a temperature between 4 °C and 6 °C (the flask containing the fluid was kept on ice), with recirculating 50 mL of Machine Perfusion Solution (MPS, modified UW; Bridge to Life) containing 3 × 10^6^ green-fluorescent MSCs, respectively. The circuit consisted of a roller pump (Masterflex) and silicone tubing with an inside diameter of 3 mm, resulting in a flow rate of 2 mL/min of perfusate to the kidney via a 24 G cannula. Kidneys were floating in a glass flask containing the MPS.

#### 4.5.2. HMP of Porcine and Human Kidneys

For kidneys undergoing HMP (n = 12 for porcine kidneys; n = 3 for human kidneys), 500 mL of Machine Perfusion Solution (MPS, Belzer (Northbrook, IL, USA)) was used for the circuit, respectively. All kidneys were perfused at a systolic pressure between 40 mmHg and 50 mmHg for 4 h. Within the porcine HMP + MSC groups, porcine kidneys were perfused with 1 × 10^6^ MSCs (n = 2), 3 × 10^6^ MSCs (n = 2) or 5 × 10^6^ (n = 2) MSCs per 500 mL of MPS. Human kidneys were equally perfused with 1 × 10^6^ (n = 1), 3 × 10^6^ (n = 1) or 5 × 10^6^ (n = 1) MSCs. MSCs were procured by trypsinization in the lab and then infused into the HMP circuit via the arterial silicone reservoir of the cassette straight after starting the 4 h HMP period. Cells were infused in a volume of 10 mL of MSC growth medium. Perfusate and kidneys were kept at temperatures between 4 °C and 6 °C. Temperature, arterial blood pressure, intrarenal resistance indices (RRIs), perfusate flow rates and urine production rates were documented at 0, 0.25, 0.5, 1, 2, 3 and 4 h of perfusion. Perfusate samples were collected at these timepoints for biochemical analysis.

#### 4.5.3. Whole-Blood Normothermic Reperfusion of Porcine Organs

After 4 h of HMP with or without MSCs, kidneys were reperfused with autologous whole blood. Simulated reperfusion on the RM3 pulsatile perfusion machine was performed for all kidneys for two hours at 36–37 °C, with a systolic perfusion pressure of 100 mmHg. Paired kidneys underwent reperfusion at the same time and on the same machine. Unpaired kidneys underwent reperfusion at different times, and separately, but were analyzed for the same parameters. Oxygen (95% O_2_/5% CO_2_) was added to the perfusate through an oxygenator at a flow rate of 0.2 L/min. Temperature, arterial blood pressure, RRI, perfusate flow rates and urine production rates were recorded at 0, 0.25, 0.5, 1 and 2 h of reperfusion. Furthermore, at 0, 0.25, 1 and 2 h of simulated reperfusion, arterial and venous blood samples were analyzed using an ABG machine. Further blood and urine samples were stored for further analysis in the lab.

### 4.6. Histological Analysis

All histology samples obtained (halved rodent kidneys and 4 mm punch biopsies from porcine and human kidneys) were fixed in 4% paraformaldehyde (PFA), followed by paraffin embedding and cutting into 4 mm sections. Dewaxed sections were covered with DAPI, and photographs were taken using widefield microscopy at 20× and 40× magnification.

### 4.7. Microscopy

Paraffin-embedded sections were analyzed using a Zeiss Axio Observer widefield inverted microscope with Zen acquisition software (https://www.zeiss.com/microscopy/en/products/software/zeiss-zen.html, accessed on 24 March 2024). Photographs were taken at 20× and 40× magnification.

### 4.8. Anti MHC-I Staining of Porcine Histology Sections

For porcine tissue samples, immunohistochemical staining against human MHC class I was performed (antibody: mouse anti Human HLA Class I Heavy Chain, Clone—HC10—Cat. Nr. MUB2037P) on dewaxed paraffin embedded slides using the DAKO rabbit envision kit, according to manual’s instructions. The antibody was used in a 1:200 dilution. Slides were counterstained in Hematoxylin and Eosin (H&E), dehydrated and mounted in Pertex before photographs were taken at 20× and 40× magnification, using light microscopy.

### 4.9. Analytical Methods

During normothermic reperfusion with whole blood, functional parameters (perfusate flow rates, RRIs and urinary output rates) were documented and compared between groups. Arterial and venous blood samples were analyzed using a blood gas analyzer to determine pH and lactate levels, and urinary and serum samples were collected for biochemical analysis (creatinine and sodium concentrations) performed by the clinical pathology lab.

Creatinine clearance was calculated to determine glomerular filtration rate (GFR) per gram of tissue as (urinary creatinine × urinary volume/creatinine in reperfusion solution). Fractional excretion of sodium was calculated as (urinary sodium × urinary flow)/(GFR × sodium in reperfusion solution) × 100/g. Oxygen consumption was calculated as (difference between arterial and venous partial pressure of oxygen) × flow rate/100 g of tissue.

### 4.10. Statistical Analysis

Statistical method values are represented as mean and standard deviation. Continuous variables were plotted as levels versus time curves for each kidney, and the mean area under the curve was calculated using ANOVA. Variables at defined timepoints were compared using Student’s *t*-test. Data and statistical analysis were performed using GraphPad Prism (Version 9). All tests were two-tailed, and a *p*-value of ≤0.05 was considered significant.

## Figures and Tables

**Figure 1 ijms-25-05038-f001:**
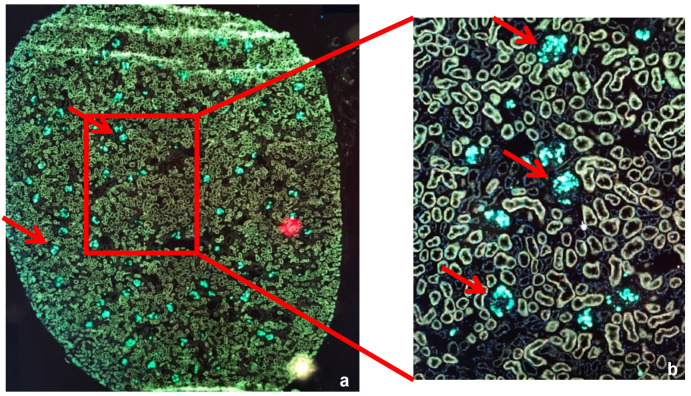
Widefield microscopy imaging of dewaxed paraffin embedded kidney sections (4 μm) with settings to stimulate the green fluorescent protein. The slides were covered with DAPI, therefore nuclei are shown in blue. (**a**) 4 mm section of a rat kidney cut in half at a 20× magnification. Mesenclymal Stem Cells (MSCs) within the glomeruli were clearly visible in bright green (red arrows). (**b**) 40× magnification of the same section, MSCs within the glomeruli were clearly visible (red arrows).

**Figure 2 ijms-25-05038-f002:**
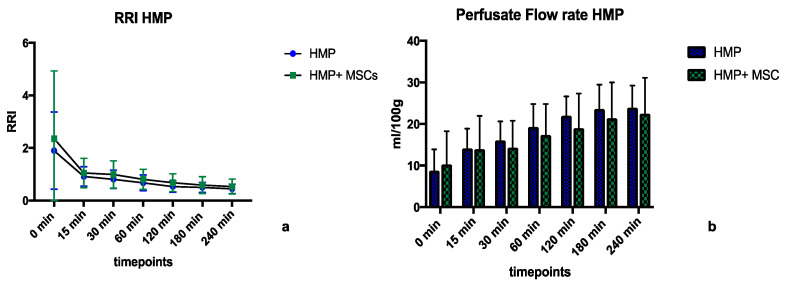
(**a**,**b**) show the intrarenal resistance index (RRI) and the resulting perfusate flow rates during the hypothermic machine perfusion (HMP) phase of 4 h, with and without Mesenchymal Stem Cells (MSCs) in the perfusate. HMP = Hypothermic Machine Perfusion, HMP + MSC = Hypothermic Machine Perfusion + 1–5 × 10^6^ double labelled MSCs in the perfusate.

**Figure 3 ijms-25-05038-f003:**
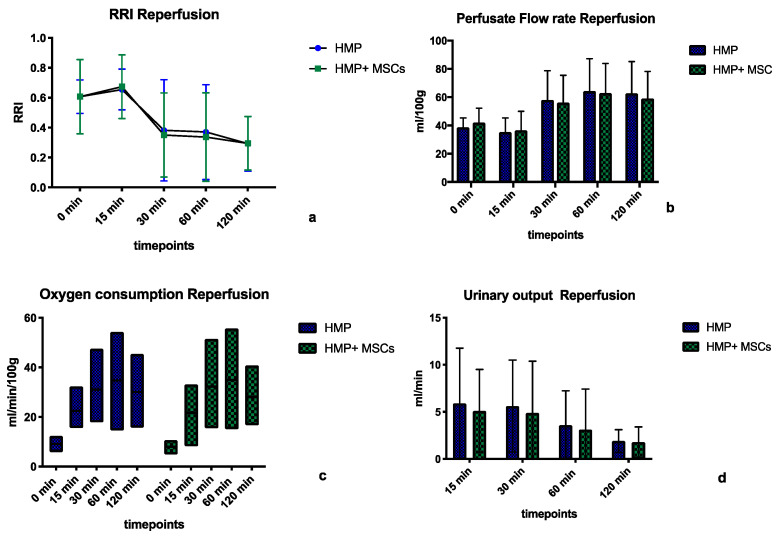
Physiological parameters of kidneys after hypothermic machine perfiusion (HMP) with or without infusion of Mesencymal Stem Cells (MSCs) during the 2 h reperfusion period. No significant difference was observed for values of intrarenal restistance indices (RRI) (**a**), perfusate fou rates (**b**), oxygen consumption (**c**) and urinary output (**d**). Statistical analyses carried out using ANOVA, statistical significance: no statistical significance (ns; no asterisk). HMP = Hypothermic Machine Perfusion, HMP + MSC = Hypothermic Machine Perfusion + 1–5 × 10^6^ double labelled MSCs in the perfusate.

**Figure 4 ijms-25-05038-f004:**
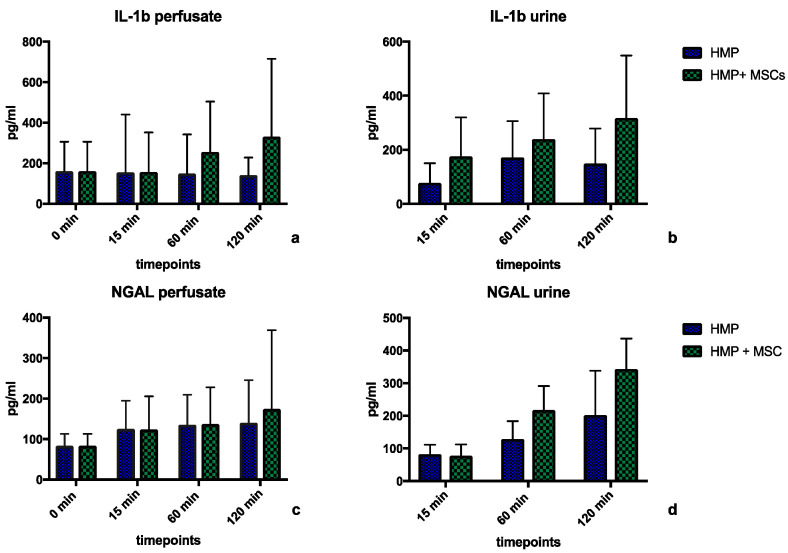
Concentrations of Interleukin-lβ (IL-Iβ) in perfusate samples (**a**) and urine samples (**b**) for porcine kidneys after 4 h of normothermic machine perfusion only (NMP), or 4 h of normothermic machine perfusion + treatment with Mesenchymal Stem Cells (NMP + MSC), measured at defined time points during reperfusion. The same measurements were made and compared for NGAL concentrations in perfusate- (**c**) and urine samples (**d**). Statistical analyses carried out using ANOVA, statistical significance: no statistical significance (ns; no asterisk). HMP = Hypothermic Machine Perfusion, HMP + MSC = Hypothermic Machine Perfusion + 1–5 × 10^6^ double labelled MSCs in the perfusate.

**Figure 5 ijms-25-05038-f005:**
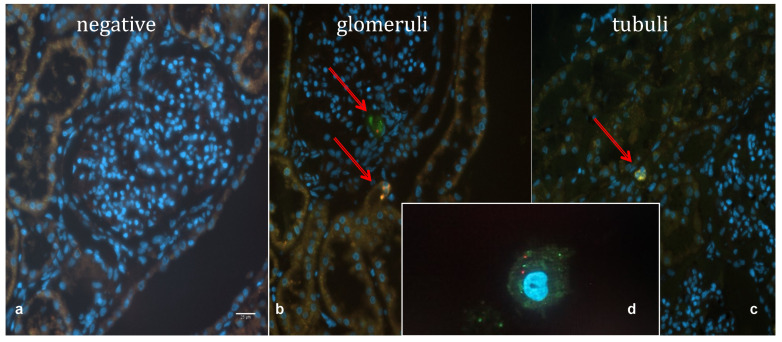
Imaging of dewaxed, paraffin embedded porcine kidney sections covered zoith DAPI (nuclei stained in bule). (**a**) shows a negative control, which was a kidney undergoing hypothermic machine perfision (HMP) only. In (**b**) 2 fluorescing cells could be visualised within one glomerulm (arrows), but cells could also be detected within the epithelium of some tubuli (**c**). In (**d**) a single doble-labelled Mesenchymal Stem Cell is shown as a direct comparison to demonstrate the similarity of the cells foitnd within the kidney. (**a**–**d**) were taken at 40× magnification.

**Figure 6 ijms-25-05038-f006:**
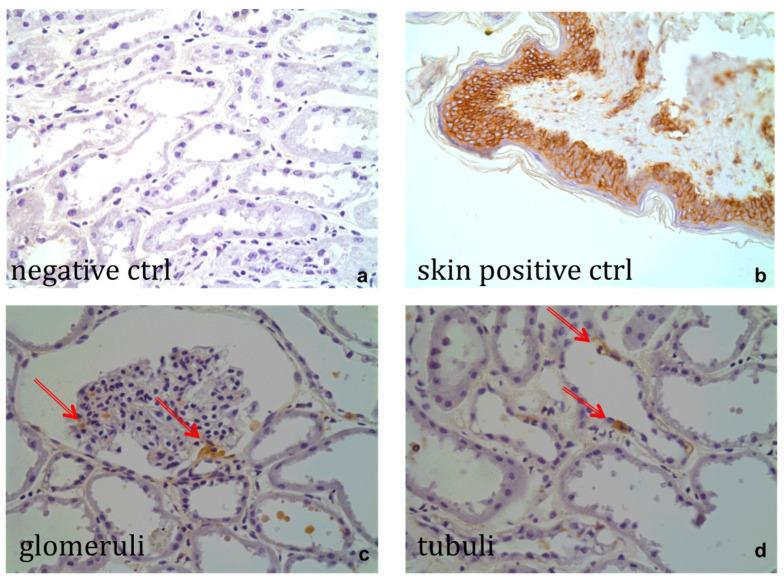
Immunohistochemical staining of porcine slides against human major histocompatibility complex-1 (MHC-1). (**a**) Negative control, (**b**) human skin sample as positive control, (**c**) positive staining for Mesenchymal Stem Cells (MSCs; arrows) within a glomerulum of a porcine kidney after 4 h of hypothermic machine perfusion (HMP) with 5 × 10^6^ MSCs, (**d**) positive staining of MSCs (arrows) within the tubuli of a porcine kidney after 4 h of HMP containing 5 × 10^6^ cells. (**a**–**d**) were taken at 40× magnification.

**Figure 7 ijms-25-05038-f007:**
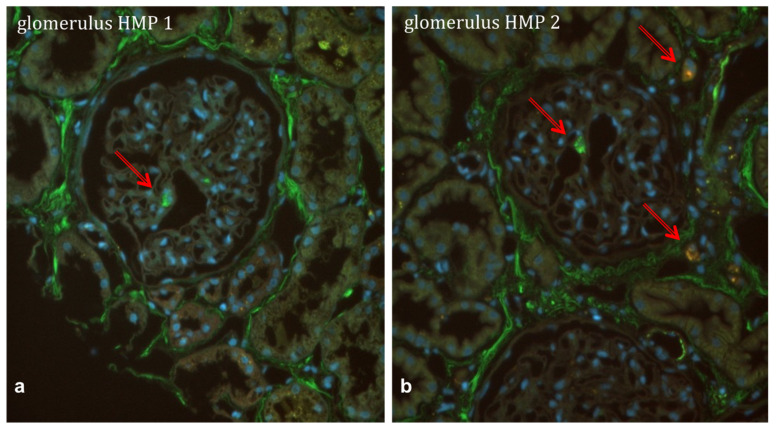
Widefield microscopy imaging of dewaxed, paraffin embedded human kidney sections (4 μm) with settings to stimulate the double labelled Mesenchymal Stem Cells (MSCs; UV and GFP). The slides were covered with DAPL, therefore nuclei are shown in blue. (**a**) picture of a fluorescent MSC within the glomerulum of a kidney after 4 h of hypothermic machine perfusion (HMP) with 5 × 10^6^ double-labelled MSCs (arrows). (**b**) another frozen section of the same kidney after HMP with 5 × 10^6^ MSCs, showing three detected cells within a glomerulum as well as within the intertubular space (arrows). (**a**,**b**) were taken at 40× magnification.

## Data Availability

Data are contained within the article.

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
