# Peer review of "Delivery of Mesenchymal Stem Cells during Hypothermic Machine Perfusion in a Translational Kidney Perfusion Study"

_ijms, 2024, doi:10.3390/ijms25095038_

Round 1

Reviewer 1 Report

Comments and Suggestions for Authors

To the Editor and the Author

The authors performed hypothermic machine perfusion on rat, pig, and human kidneys and confirmed that administration of labeled MSCs in the circuit allowed the cells to remain in the organ.

This topic of organ conditioning with MSCs during machine perfusion is one that many researchers are focusing on, and it is meaningful.

The photograph in which the administered MSCs appear to be integrated into porcine renal tubules was impressive.

However, many shortcomings divert this manuscript away from the level of

publication. Some crucial matters are described lazily, reducing the credibility of the authors' claimed results and the paper itself.

Major points

1. Method is insufficient and difficult to understand.

Donor animal information is lacking. e.g., weight of rats, sex of rats and pigs.

It should be noted that the left and right kidneys of the pigs were paired and

assigned to the treatment and control groups.

There is also a lack of description of the assignment of animals with different

numbers of cells administered.

The authors should also state which animal MSCs were administered to which

animal kidneys (especially regarding administering human cells to pigs). Some of these are mentioned in the Discussion, but the Method should be completed by itself.

2. Regarding the endpoints, the degree of IRI after pig whole blood reperfusion should be evaluated compared to the control.

3. Figures 1 and 4 should include stronger magnification to show where the

cells are located in the glomerulus. Fig5 should also be stained for vascular

endothelium, mesangium, podocytes, etc., to see where the MSCs are located

in the glomerulus.

4. Why did the authors administer human cells to pigs? Couldn't they have created and labeled pig MSCs? Can cell adhesion really occur between cells of different species, and can the xenogeneic MSCs be integrated into the tubules?

5. Figs 7-10 appeared in the text, but the figures are missing.

6. IL-β is highlighted in the abstract as the only outcome that changed; the

Discussion should include the meaning of IL-β in this context and the expected mechanism of this change、

7. In page 8, lines 234-237, on what basis do the authors claim that the

administered cells are Viable? Were dead cells with apoptosis or lyse actually

observed?

Minor points

1. In page 3, line 90, The authors state that "we did not observe any differences in regards to perfusate flow rates or RRI." The RRI appears to be slightly higher and the flow rate appears to be slightly lower. Since the Small sample size makes it not definitive, please change the wording to a toned-down version.

2. In Result 2.2.3., a graph should be shown for the results of IL-beta since it is the only outcome that showed a difference.

3. Fig6a states that double-stained cells are present, but they do not appear so.

4. In page 7, line 201, if the authors insist on safety, they should conduct

transplantation experiments, even if only for a short period.

5. Many words appear as abbreviations without explanation ( e.g., RRI, CSS,

MPS, WBNR, etc.).

6. Method 4.6 states that all samples were processed as frozen sections, but

Method 4.8 states that paraffin sections were processed.

7. The latter half of Method 4.4 and the latter half of 4.5.3 are duplicated; I

understand that it is difficult to describe the complicated experimental system using three different animal species, but the structure should be designed to be easily understood by the readers.

Comments on the Quality of English Language

English in this paper: Moderate editing of English language is required.

Author Response

Dear reviewer, thank you very much for the thorough review and for the most helpful comments. We hope we have addressed all of them sufficiently, please see the point by point answers below:

Major points:

  1. Weights and genders of animals have now been included.
  2. We agree that the degree of IRI is relevant in regards to the application of MSCs in ex-vivo HMP. However, this was not the focus of this specific paper. At first, we wanted to evaluate whether it was feasible and safe to perfuse MSCs into kidneys in a HMP setting and whether those MSCs could be delivered into grafts.  During 2 hours of reperfusion, we would not yet expect differences in regards to IRI being visible. 
  3. This is a good point and would be great to do. Unfortunately this is not possible to do retrospectively. 
  4. Thank you for this valid point. Unfortunately our attempt at extracting MSCs from a porcine femur failed. We were facing the problem of contamination of the culture with a fungus. Commercially available porcine MSCs did not exist at the time. Hence, we looked up the literature and found evidence that porcine MSCs could be used in this context. Please see as example, the following paper:  Noort WA, Oerlemans MI, Rozemuller H, Feyen D, Jaksani S, Stecher D, Naaijkens B, Martens AC, Bühring HJ, Doevendans PA, Sluijter JP. Human versus porcine mesenchymal stromal cells: phenotype, differentiation potential, immunomodulation and cardiac improvement after transplantation. J Cell Mol Med. 2012 Aug;16(8):1827-39. doi: 10.1111/j.1582-4934.2011.01455.x. PMID: 21973026; PMCID: PMC3822. 
  5. Done. Thank you for the valuable comment. 
  6. Will address
  7. Most likely, cells which are still looking round and contain cell stain would be alive. Otherwise most likely the cell would have gone into apoptosis and lysed and hence, would not be visible. There is also no reason why the cells would not be viable when kept in an organ preservation solution. 

Minor points:

  1. We have stated that the RRIs and perfusate flow rates were similar in both groups. We believe that is the correct term to describe these numbers.
  2. Done. Thank you very much
  3. It is hard to depict what can be seen in much higher resolution in the actual microscopy room at the FILM facility at Imperial College London. Together with experts who only focus on microscopy, we identified the MSCs within those kidneys. The fact that the microscopy experts had nothing to do with our research project was very reassuring because they did not know the meaning of telling us where the cells were. Hence, the only way to figure out the location of the cells during this was to trust those researchers pointing the cells out to us. 
  4. This will be the next step. In the ex vivo studies however, nothing points towards the cells not being safe for administration. It is also known from literature that it is safe to apply the cells. 
  5. This has been addressed and changed.
  6. Thank you, changes were made accordingly.
  7. Changed. Many thanks. 

Reviewer 2 Report

Comments and Suggestions for Authors

Dear authors, thank you very much for giving me the opportunity to read your work. You are presenting data proving the feasibility of the ex vivo mesenchymal stem cell infusion and engraftement into the kidneys. Your work is novel. 

I have the following questions/comments: 

Human/Porcrine kidneys:

Did you experiment with stem cells originating from 1) different donors, 2) donors having different ages? Was there any difference in the number of migrating stem cells? 

Il-1β, appart from its secreted form, has a membrane bound fraction. Is there evidence of the membrane bound Il-1β fraction on stem cells before infusion? Is there any observed change when incubating the stem cells with the perfusion medium?

Did you study other markers of tubular stress? For example KIM1, DKK3? What is the rationale for solely choosing  NGAL?  

All of my best regards.

Author Response

Dear editor,

Many thanks for reading our work and reviewing so promptly and thank you for the most stimulating comments and question. 

To answer them point by point:

1.) In regards to the Mesenchymal Stem Cells, we were a bit limited as to the source for them. We were kindly given a vial of human Mesenchymal Stem Cells by our colleagues from King's College London, who have ethical approval for the use of those human cells for research. We then expanded the cells in culture and had to maintain this very culture for our experiments. We attempted to extract Mesenchymal Stem Cells from a pig's femur. Unfortunately this did not succeed. We then found evidence in the literature that Mesenchymal Stem Cells from humans and pigs are very similar in their phenotype, please see for example the following paper:

Noort WA, Oerlemans MI, Rozemuller H, Feyen D, Jaksani S, Stecher D, Naaijkens B, Martens AC, Bühring HJ, Doevendans PA, Sluijter JP. Human versus porcine mesenchymal stromal cells: phenotype, differentiation potential, immunomodulation and cardiac improvement after transplantation. J Cell Mol Med. 2012 Aug;16(8):1827-39. doi: 10.1111/j.1582-4934.2011.01455.x. PMID: 21973026; PMCID: PMC3822. 

We do indeed agree however, that MSCs from different donors of different backgrounds, for example age, medical background etc., might be different in their actions. This will indeed need to be explored. However, realistically, in a transplant setting, the idea would be to optimise grafts with the recipient's own MSCs. 

2.) For the human MSCs, we did not perform any in vitro experiments. This would indeed be something very interesting to look at. Many thanks for the useful idea and comment. 

3.) Thank you for the very helpful question. We chose to look at NGAL as one of the best known markers for kidney injury. We know that NGAL has also been used to monitor kidney injury in clinical settings and therefore, by choosing this marker, we think that translating our work into clinical practice might be facilitated. We have also looked at markers like Endothelin 1 on and TNFalpha on mRNA level, but found no significant differences in the expression between the groups, hence that data was not shown. 

We hope this answers your questions sufficiently, the paper has been updated and improved thanks to your constructive comments. 

Many thanks, 

Best wishes.

Round 2

Reviewer 1 Report

Comments and Suggestions for Authors

The authors made only slight revisions, and the overall impression of this manuscript is the same as the previous one.

The immunostaining images and their interpretation, which is the key of this paper, are inaccurate and there are important questions about the methodology. (Despite my pointing this out last time, the paper still states that the frozen section was used in the text and the Paraffin section was used in the methodology.).

I feel that the results of this paper and the claims made by the authors are unreliable.

Comments on the Quality of English Language

Moderate editing of English language is required.

Author Response

Dear Reviewer,

Many thanks for your comments.

In regards to the English used, I am not sure how to improve this. I would have hoped having lived in the UK for the past 10 years would have given me sufficient skills to write in proper English. Apologies if the English isn't at the highest level possible. 

In regards to the revisions, we have made all the revisions as far as they were possible on the paper. Unfortunately some of the points were simply not doable as we cannot go back in time to perform the research again. 

We do understand the concerns about the methodology. For this project, a translational model, a lot of methods were required. To put them down in writing is a challenge. We would not know how to make it any clearer. 

In regards to the frozen sections, this was a sincere mistake, which was made simply because the initial plan was to use frozen sections. We were then advised by our Histopathologists who processed the slides for us, that paraffin embedded slides would equally be adequate. This made the methodology slightly easier for the actual researcher. Allover, this would not change the outcome. We are however appreciative of the fact you kindly highlighted the mistake so we could change it. 

With many thanks and kind wishes